# Correlating Josephson supercurrents and Shiba states in quantum spins unconventionally coupled to superconductors

Felix Küster[1,4], Ana M. Montero [2,4], Filipe S. M. Guimarães [2], Sascha Brinker [2], Samir Lounis [2,3✉], Stuart S. P. Parkin [1✉] & Paolo Sessi [1✉]

Local spins coupled to superconductors give rise to several emerging phenomena directly linked to the competition between Cooper pair formation and magnetic exchange. These effects are generally scrutinized using a spectroscopic approach which relies on detecting the in-gap bound modes arising from Cooper pair breaking, the so-called Yu-Shiba-Rusinov (YSR) states. However, the impact of local magnetic impurities on the superconducting order parameter remains largely unexplored. Here, we use scanning Josephson spectroscopy to directly visualize the effect of magnetic perturbations on Cooper pair tunneling between superconducting electrodes at the atomic scale. By increasing the magnetic impurity orbital occupation by adding one electron at a time, we reveal the existence of a direct correlation between Josephson supercurrent suppression and YSR states. Moreover, in the metallic regime, we detect zero bias anomalies which break the existing framework based on competing Kondo and Cooper pair singlet formation mechanisms. Based on first-principle calculations, these results are rationalized in terms of unconventional spin-excitations induced by the finite magnetic anisotropy energy. Our findings have far reaching implications for phenomena that rely on the interplay between quantum spins and superconductivity.

[1] Max Planck Institute of Microstructure Physics, Halle, Germany. [2] Peter Grünberg Institut and Institute for Advanced Simulation, Forschungszentrum Jülich & JARA, Jülich, Germany. [3] Faculty of Physics, University of Duisburg-Essen, Duisburg, Germany. [4] These authors contributed equally: Felix Küster, Ana M. Montero. ✉email: s.lounis@fz-juelich.de; stuart.parkin@mpi-halle.mpg.de; paolo.sessi@mpi-halle.mpg.de

The competition between magnetism and superconductivity is one of the most fascinating, highly debated, and intriguing topics in condensed matter physics. After the formulation of the BCS theory[1], it became clear that superconductivity in the spin-singlet state is destroyed by a magnetic exchange mechanism which tends to align the opposite spins of Cooper pairs in the same direction, thus preventing their formation, i.e., the so-called para-magnetic effect[2,3]. Consistently with theoretical expectations, early experimental works using heat capacity, transport, and tunneling junctions measurements evidenced a reduction of the super-conducting transition temperature when magnetic impurities were introduced into the system[4–8]. However, by averaging over the entire sample's area, these techniques rely on the assumption of equivalent impurities, inevitably including spurious effects related to sample inhomogeneity or contaminants. Overall, this severely complicated the task of disentangling the role of spin from that of the local environment. This shortcoming has been overcome by the invention of experimental methods capable of capturing the rich physics taken place at the nanoscale by atomic resolution imaging[9]. In a seminal scanning tunneling microscopy (STM) work, Yazdani et al. visualized the effect of single magnetic impurities coupled to an elemental superconductor, demonstrating the presence of an enhanced density of states residing inside the superconducting energy gap[10]. By using a classical spin model, these results were explained in terms of magnetic exchange-induced quasi-particle resonances, i.e., the so-called Yu–Shiba–Rusinov (YSR) states[11–13]. In recent years, tremendous progress has been made in under-standing YSR excitations[14–20]. These efforts were mainly driven by the identification of superconducting–magnetic interfaces as viable routes towards the creation of topological superconductors sup-porting Majorana modes[21,22], which are essential ingredients for topological quantum computation schemes[23,24]. This progress was made possible by the development of routinely available low-temperature STM-based spectroscopic techniques with an energy resolution well below the meV range which allowed one to precisely identify YSR resonances and directly link them to the single-impurity ground state[17].

However, previous studies suffer from two main limitations, namely: the inability to directly access the effect of magnetic perturbations on the superconducting order parameter and the focus on single specific perturbations, an approach that impedes the discovery of well-defined trends and correlations. Here, we overcome these limitations by (i) systematically spanning the 3d orbital occupation adding one electron at a time, and (ii) scruti-nizing the impact of each impurity in three different spectroscopic regimes: Shiba, Josephson, and metallic. Scanning Josephson spectroscopy measurements are used to directly map the effect of magnetic impurities by visualizing the suppression they induce on Cooper pairs tunneling between superconducting electrodes[25–28]. This allows to discover the existence a direct correlation between Cooper pairs tunneling and Shiba states, revealing a stronger suppression of the Josephson supercurrent for impurities hosting multiple YSR within the energy gap, an effect directly linked to their higher spin state. In agreement with ab initio calculations, this correlation is directly linked to the existence of an orbital occupation-dependent oscillatory behavior, with vanishing mag-netic interactions for elements at the opposite extremes of the 3d element series. Moreover, by driving the system in the normal metallic regime, we reveal the emergence of zero-bias anomalies which, in sharp contrast to expectations, become progressively stronger by approaching the quantum phase transition from the Kondo to the free-spin regime in the well-known phase diagram of magnetic impurities coupled to superconductors[29]. Supported by ab initio calculations based on density functional theory (DFT), relativistic time-dependent DFT (TD-DFT)[30–32], and many-body perturbation theory (MBPT)[33,34], these low-energy spectroscopic features are identified as unconventional spin excitations emerging from finite magnetic anisotropy energy.

Overall, our results shed new light on how local spins interact with superconducting condensates. They provide a self-consistent experimental picture allowing the discovery of new effects and the visualization of new trends that always escaped experimental detection so far and with far-reaching implications especially within the realm of engineered topological superconductivity.

## Results

**Experimental lineup**. The experimental lineup used to scrutinize the aforementioned aspects is schematically illustrated in Fig. 1. Local spins coupled to an electron bath are characterized by a magnetic exchange term $JS$ with $J$ being the s–d exchange cou-pling of the localized spin of the impurity $S$, carried here by d-electrons, and the conduction electrons of the substrate. Its effects are expected to manifest in three distinct ways, schematically illustrated in panels a–c. In the superconducting regime, it represents a scattering potential breaking Cooper pairs and giving rise to in-gap YSR states (Fig. 1a). In addition, it is expected to directly affect the superconducting order parameter by suppres-sing the strength of the pairing interaction, resulting in a reduction of the Josephson current flowing between super-conducting electrodes (Fig. 1b). Finally, a strong coupling between magnetic impurities and the electron bath can open additional tunneling channels. These result from inelastic spin excitations induced by the magnetic anisotropy, which opens a gap in the spectra and are experimentally signaled by a step-like increase in the experimentally detected local density of states (LDOS), as sketched in (Fig. 1c)[35]. As described in the following, instead of the usual two steps expected at positive bias and negative bias voltage, the inelastic spectra can display an unconventional shape, in accordance with recent predictions[33,34].

Figure 1d illustrates the portion of the periodic table of the 3d elements investigated in this study. By scrutinizing the 3d occupation scenario adding one electron at a time, it is possible to analyze the role of orbital occupation in determining the magnetic impurity–superconductor interaction strength. As superconducting material, we choose niobium single crystals which have been prepared according to the procedure described in ref. [36]. Niobium represents an optimal choice compared to other superconductors such as Pb[14,15], Re[37,38], and Ta[18] used in previous studies. Indeed, by having the highest transition temperature ($T = 9.2$ K) among all elemental superconductors, it allows to clearly disentangle in-gap states from superconducting gap thermal broadening effects. Panel e shows a topographic image where different magnetic impurities (Fe and Cr) have been deposited onto the clean Nb(110) surface prepared according to the procedure described in "Methods" and Supplementary Fig. 1. The very same approach has been used for all atomic species, i.e., V, Cr, Mn, Fe, and Co (see Supplementary Fig. 2 for the determination of the adsorption sites). To investigate their impact on the superconducting condensate, full spectroscopic maps have been acquired at temperature $T = 1.9$ K using superconducting Nb tips. Compared to conventional metallic tips, their use brings two crucial advantages: (i) they allow to enhance the energy resolution while simultaneously (ii) opening the fascinating possibility to measure the Josephson effect at the atomic scale.

**YSR spectroscopy**. Figure 2 reports the spectroscopic character-ization of the superconducting gap obtained by positioning the tip directly on top of the different magnetic perturbations. As described in Supplementary Fig. 3, the use of superconducting tips shifts the "zero energy" by $\pm \Delta_{tip}$ with respect to the Fermi level, $\Delta$ being the superconducting energy gap. Hence, the single-

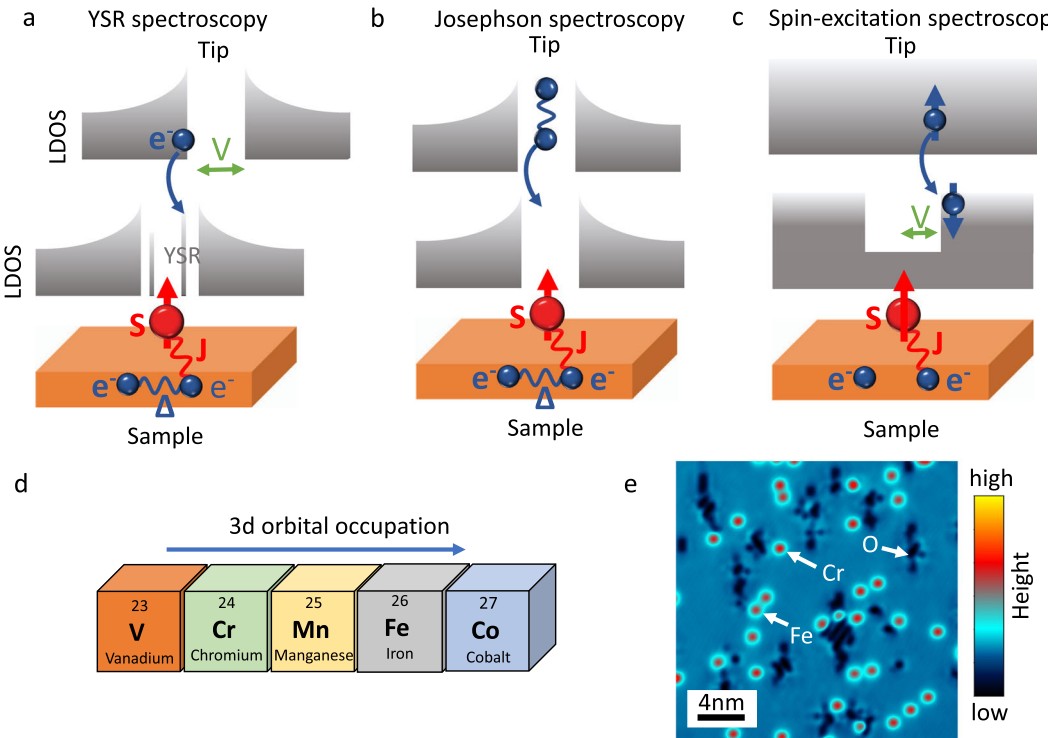

**Fig. 1 Experimental lineup.** Schematic illustration of (**a**) Yu–Shiba–Rusinov (YSR), (**b**) Josephson, and (**c**) spin-excitation spectroscopy. These three different spectroscopic modes can be used to provide a complete spectroscopic characterization of spin-related effects in the superconducting as well as in the metallic regimes. $V$ corresponds to the bias applied across the tunneling junction, $J$ indicates the exchange coupling of the localized spin of the impurity $S$, while $\Delta$ is the superconducting energy gap of electrons $e^-$ condensing into Cooper pairs, (**d**) 3d elements scrutinized in the present study, (**e**) topographic image showing Cr and Fe single atoms coupled to the Nb(110) surface. Scanning set-point: $V = -300$ meV, $I = 300$ pA.

particle coherence peak appears at energies $\pm(\Delta_{tip} + \Delta_{sample})$. In this case, this corresponds to ~±3 meV, with slight variations resulting from tips characterized by different Nb clusters at their apex (see Supplementary Fig. 4). An inspection overview of Fig. 2 allows us to immediately identify the existence of an oscillatory behavior that directly correlates with the filling of the 3d orbitals. In particular, V and Co do not show any impact on the superconducting properties, with scanning tunneling spectroscopy spectra taken by positioning the tip over the adatoms (orange line) perfectly overlapping, within our experimental resolution, the spectra taken over the bare Nb substrate (black dashed line). Despite no difference in the d$I$/d$U$ curves, a very weak $d_{z^2}$-derived YSR state can be detected for V, which is energetically overlapping with the single-particle coherence peak at the edge of the superconducting gap. These results suggest a very small and vanishing magnetic moment for both V and Co, respectively which are located at the opposite extremes of the 3d orbital scenario analyzed in the present study. Both elements are characterized by a partially filled 3d shell, this behavior might appear surprising and it highlights how the hybridization with the substrate can dramatically impact the magnetic properties. Similar to our finding, Co adatoms can be nonmagnetic on Re surface as revealed by a YSR study limited to Mn, Fe, and Co impurities[38]. As described in the following, the trend unveiled by our experiments is confirmed by ab initio calculations (see subsection "Ab initio simulations").

In contrast, well-defined YSR states emerging within the superconducting gap are visible for Cr, Mn, and Fe. As expected, all YSR states appear in pairs symmetrically located around the Fermi level. Their energy position $\epsilon$ within the superconducting gap is generally described considering pure magnetic scattering mechanisms, being determined by the strength of the exchange

coupling terms $J$ through the following expression:

$$\epsilon = \pm\Delta\frac{1-\alpha^2}{1+\alpha^2}$$

with $\alpha = \pi\rho J S$, $S$ being the impurity's spin, and $\rho$ the sample density of states at the Fermi level in the normal state[20]. For each pair, the different intensities between occupied and unoccupied resonances can be used to identify whether the YSR state is in a screened-spin (higher intensity for hole injection, i.e., $E < E_F$) or a free-spin configuration (higher intensity for electron injection, i.e., $E > E_F$)[16,20].

In the case of Fe, a single pair of YSR states is detected. It energetically overlaps with the single-particle coherence peaks visible at the edge of the superconducting energy gap. Spatially mapping its intensity allows one to assign it to a $d_{z^2}$ scattering orbital (see colormaps in Fig. 2d). Cr and Mn show a more complicated spectrum supporting multiple YSR pairs. As for Fe, a $d_{z^2}$ scattering orbital is clearly visible, which moves towards smaller binding energies by progressively decreasing the atomic number. The additional YSR pairs are located at different energies within the superconducting gap. Their spatial distribution is far from being isotropic, resembling well-defined d-level symmetries. These observations prove that the magnetic exchange scattering potentials are strongly orbital-dependent[17]. Interestingly, in Fig. 2c, the Mn $d_{xz}$-derived Shiba pair show distinct spectral maps at positive and negative energies, signaling a strong particle–hole asymmetry in the wavefunctions, similarly to $d_{z^2}$ YSR-bound states. An additional pair is visible at energies $\pm(\Delta_{tip} - \epsilon)$. These states correspond to the thermal replica of the $d_{xz}$-derived Shiba pair: they become populated by particles and holes due to their proximity to the Fermi level. This assignment is further confirmed by their shape, which energetically mirrors that of the original states[17].

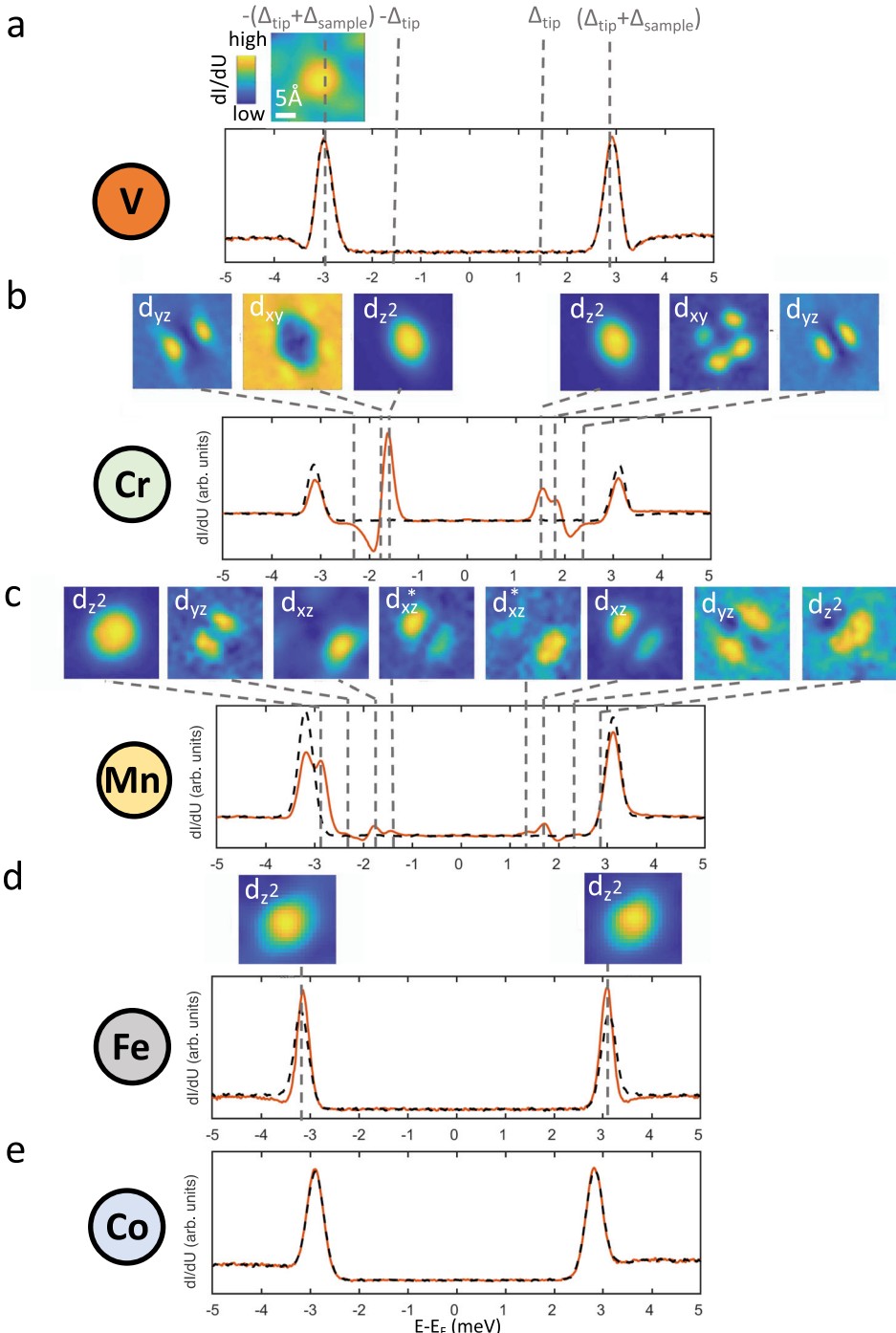

**Fig. 2 Yu–Shiba–Rusinov (YSR) spectroscopy. a–e** Scanning tunneling spectroscopy for V, Cr, Mn, Fe, and Co. While no YSR state is detected for V and Co, a rich set of in-gap states emerge for Cr, Mn, and Fe. The insets show the spatial distribution of the YSR states, which reflect their orbital character and allows one to quantify their spatial extension. Stabilization parameters: $V = 5$ meV, $I = 500$ pA.

Interestingly, these results allow to systematically follow the evolution of the $d_{z^2}$-derived YSR state, visualizing how it progressively moves toward higher binding energies by increasing the $3d$ orbital occupation. Within the generally assumed framework of competing singlet-formation mechanism, i.e., Kondo vs. Cooper pairs, this is expected to result in Kondo resonances becoming progressively stronger by moving from Cr to Mn, and, Fe. However, as demonstrated in the following, this is far from being the case (see section "Spin excitations and related discussion").

**Josephson spectroscopy**. Although YSR measurements can be effectively used to infer important information on the magnetic coupling strength, they are characterized by a strong fundamental limitation: they can not visualize the effect of magnetic impurities on the superconducting order parameter. Indeed, the local pairing suppression which is expected to take place in presence of magnetic perturbation can not be directly reflected in the YSR spectra. As illustrated in Fig. 2, these show a suppression in the intensity of the coherence peaks at the edge of the superconducting gap, their spectral weight being redistributed to the in-gap bound

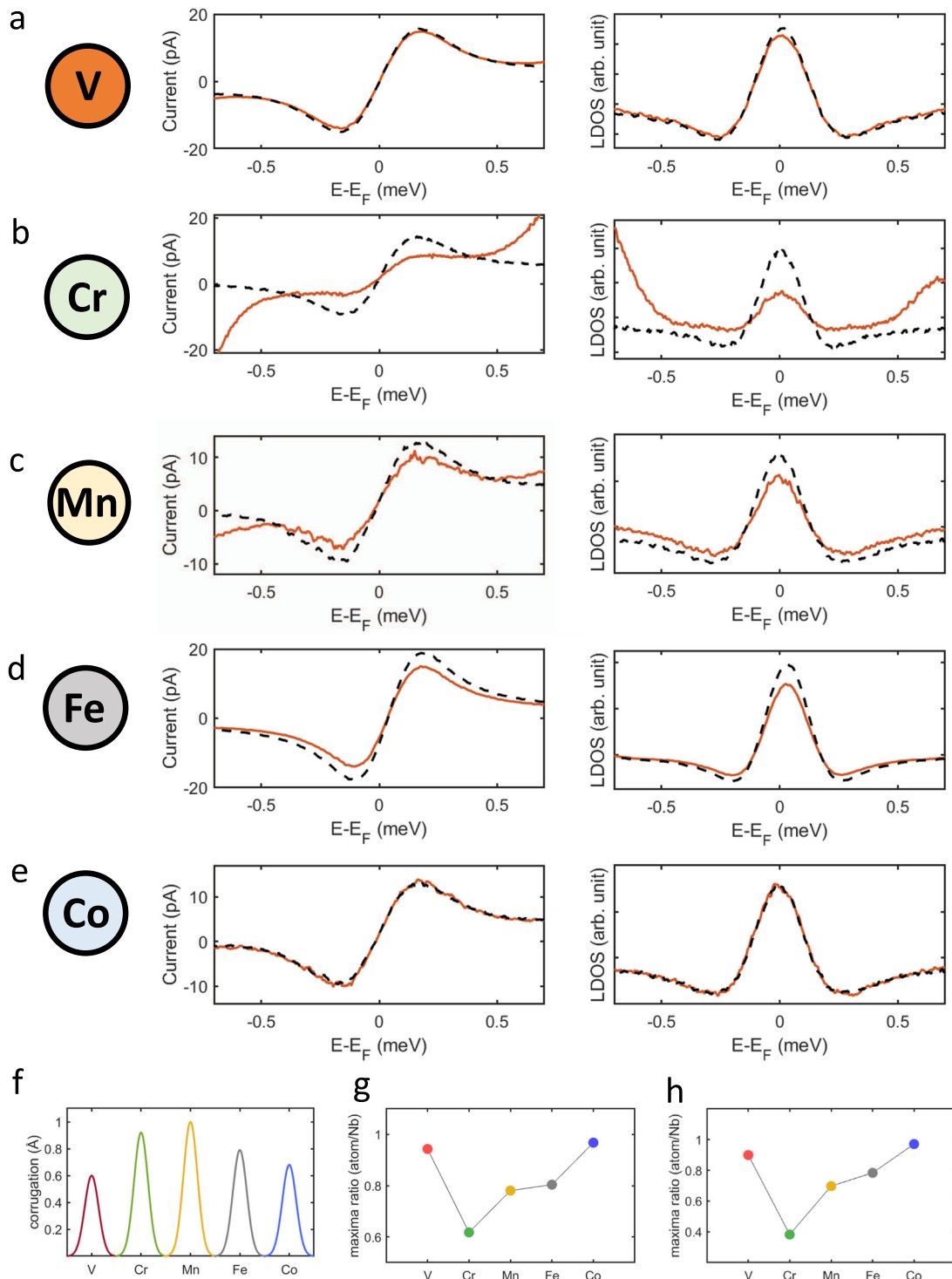

**Fig. 3 Scanning Josephson spectroscopy. a–e** $I - V$ characteristics (left panels) and respective $dI/dU$ signals (right panels) for all elements. **f** Apparent heights of the adatoms. **g**, **h** Reduction induced by the different magnetic perturbations in the maximum Cooper pairs current $I_{max}$ and intensity of the $dI/dU$ peak, respectively, both showing the same trend. Stabilization parameters: $V = 5$ meV, $I = 15$ nA.

states, but without any energy shift of their position as compared to the substrate. This distinction between detecting the effects of magnetic impurities on the local density of states and on the superconducting order parameter is well-known and consistent with theoretical expectations[39].

To overcome this limitation, we perform scanning Josephson spectroscopy measurements which allow, by measuring the

tunneling between Cooper pairs in superconducting electrodes, to directly extract information on the local variation of the superconducting pairing amplitude at the atomic scale. Results for all investigated impurities are summarized in Fig. 3, with left panels showing the $I$–$V$ characteristics and the right ones reporting their respective $dI/dU$ signal. The orange and dashed black lines have been acquired by positioning the tip atop the

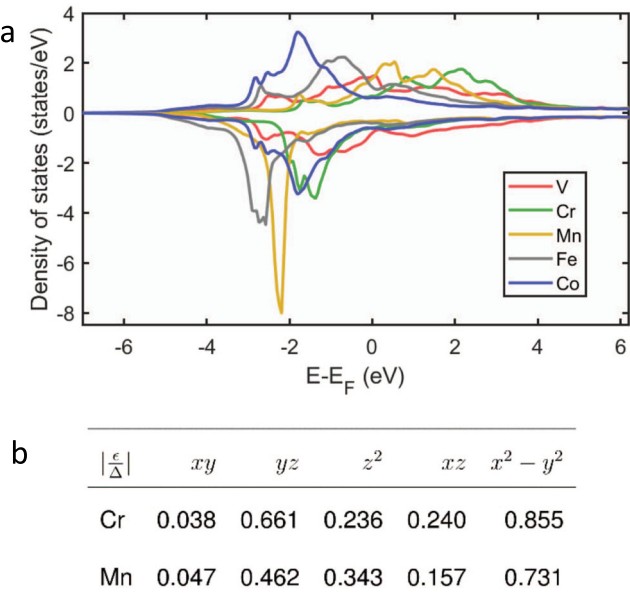

**b**

| $\left|\frac{\epsilon}{\Delta}\right|$ | $xy$ | $yz$ | $z^2$ | $xz$ | $x^2-y^2$ |
|---|---|---|---|---|---|
| Cr | 0.038 | 0.661 | 0.236 | 0.240 | 0.855 |
| Mn | 0.047 | 0.462 | 0.343 | 0.157 | 0.731 |

**Fig. 4 Local density of states and energies of YSR states. a** Spin-resolved electronic structure for V, Cr, Mn, Fe, and Co—upper (lower) panel for minority- (majority-) spin channel. **b** Theoretically obtained table of orbital-dependent energies of YSR states normalized by the superconducting gap.

magnetic impurities and on the bare substrate, respectively. While no difference with respect to the bare Nb substrate is detected for Co, a small reduction of the Josephson current is observed for V. This effect becomes stronger for all other impurities according to the following order: Fe, Mn, and Cr. These results confirm the oscillatory magnetic behavior visualized in Fig. 2: the magnetic moment becomes strongly suppressed at the opposite extreme of the 3$d$ filling, while Fe, Mn, and Cr preserve a substantial magnetic character. The effect of the different magnetic impurities on the superconducting order parameter can be quantitatively analyzed based on the fact that both the maximum Cooper pairs current $I_{max}$ as well as the intensity of the d$I$/d$U$ peak are proportional to the square of the intrinsic Josephson current $I_c$[27]. A direct comparison among the different adatoms evidences a suppression of the Cooper pairs tunneling, an effect directly linked to the reduction of the superconducting order parameter, which becomes progressively stronger by moving from Fe to Mn and, finally, Cr. In agreement with theoretical expectations, the same trend is consistently found for both $I_{max}$ (Fig. 3g) and the d$I$/d$U$ peak intensity (Fig. 3h)[27]. Although different magnetic impurities have a different apparent height, as shown in panel e, we can safely exclude that our observations are related to tip-height artifacts. Indeed, despite having Co an apparent height of ~0.7 Å, Josephson currents are not altered when compared to the case when the tip is positioned over the bare substrate. Furthermore, Cr, which shows the strongest perturbation on the superconducting order parameter, has an apparent height that is smaller than Mn. Additional experimental evidence ruling out tip-height effects is provided in Supplementary Fig. 5 where, by using atomic manipulation techniques, we create a Cr dimer. Although being apparently higher than a single Cr adatom, the dimer does not have any impact on the superconducting order parameter, an observation consistent with its antiferromagnetic ground state resulting in a total spin $S = 0$. Consequently, our measurements directly fingerprint effects induced by a finite spin onto the superconducting order parameter, suggesting a progressively increasing magnetic moment while moving from Fe to Mn and finally Cr. As

discussed in the following, these results follow the same trend of the magnetic moments obtained by our theoretical calculations and highlight the very high sensitivity of our measurement protocol.

**Ab initio simulations.** The theoretical interpretation of the trends observed in both YSR and Josephson spectra requires detailed knowledge of the spin-resolved orbital structure of the adatoms and their coupling to the substrate. This is analyzed in the following on the basis of ab initio simulations of the 3$d$ series of adatoms deposited on Nb(110) surface (see Supplementary Notes 1–3 for more details). Figure 4a reports the spin-resolved local density of states (LDOS) for V, Cr, Mn, Fe, and Co with upper and lower panels corresponding to minority- and majority-spin channels, respectively. The LDOS broadening is a direct consequence of the crystal field, which splits the degeneracy of the different 3$d$ orbitals. A detailed discussion is provided in Supplementary Notes 1–3. Its inspection immediately reveals the appearance of a well-defined trend: a substantial imbalance between majority- and minority-spin resonances is found for Cr, Mn, and Fe, while the difference between majority- and minority-spins is found negligible for V and totally absent for Co. These results follow the usual inverse parabolic behavior across the 3$d$ series, with spin magnetic moments reaching a maximum in the middle followed by a decrease toward the end of the series. In agreement with our experimental observations, only four adatoms remain magnetic, with elements at half-filling of the $d$-states carrying the largest moments (V: ~1.2 $\mu_B$; Cr: ~3.5 $\mu_B$; Mn: ~3.6 $\mu_B$; Fe: ~2.0 $\mu_B$) while Co is nonmagnetic. Note that a non-negligible magnetic moment is induced in the bare Nb substrate at the vicinity of the adatoms, to which it generally couples antiferromagnetically, except for V. This effect modifies the total adatoms–substrate complex spin moments, resulting in V: ~1.4 $\mu_B$, Cr: ~3.3 $\mu_B$, Mn: ~3.0 $\mu_B$, and Fe: ~1.5 $\mu_B$. These values correlate well with the trend visualized by Josephson spectroscopy measurements reported in Fig. 2, allowing to establish a direct link between the magnitude of the magnetic moment and the induced suppression of Cooper pairs supercurrents.

The strength of the orbital-average impurity-substrate hybridization, Γ, between adatoms and substrate is rather large for all the adatoms, and it decreases by increasing the 3$d$ orbital occupation, i.e., by moving from left to right across the 3$d$ series (V:1.11 eV; Cr: 0.98 eV; Mn: 0.88 eV; Fe: 0.72 eV; Co: 0.57 eV). This trend is related to the contraction of the 3$d$ states of the atoms when increasing their atomic number, which disfavors hybridization with neighboring atoms. While the hybridization strength is paramount for the description of YSR-bound states, it is worth stressing that its effect can be counteracted by the exchange splitting, 2$U$, and the energy of orbital $m$, $E_m$. A full ab initio description of the YSR states is currently challenging. Here, we follow a simplified model where the aforementioned quantities encode the magnitude of the orbital- and spin-dependent impurity-substrate $s$–$d$ interaction $\mathcal{I}_m^\sigma$, where $\sigma = \pm$ depending on the spin of conducting electrons. By virtue of the Schrieffer-Wolff transformation[40], $\mathcal{I}_m^\sigma = (V_m + \sigma J_m S)$, with $V_m$ and $J_m$ corresponding to nonmagnetic and magnetic scattering contributions, respectively. The energies of the YSR states can then elegantly be cast into[13,41]:

$$\frac{\epsilon_m}{\Delta} = \pm \cos(\delta_m^+ - \delta_m^-), \qquad (1)$$

where the phase shifts are given by $\tan \delta_m^\sigma = \pi\rho \, \mathcal{I}_m^\sigma$.

This approach is capable of mapping the scattering phase shifts and the YSR energies directly from our ab initio results (see Supplementary Notes 2 and 3). The complexity of the problem is directly related to the very different energy scales coming into

play: the interactions $J$ and $V$ depend on quantities of the eV range, while the energies of the YSR states are of the order of meV and sub-meV. This impedes a perfect one-to-one comparison between all the theoretically calculated and experimentally measured spectra. However, our approach is effectively capable of capturing the observed experimental trends, as discussed in the following. The theoretically predicted energy position for Cr and Mn YSR states is summarized in Fig. 4b.

The Cr $d_{z^2}$ Shiba state is predicted to be located at lower energy than that of Mn, in agreement with the experimental data (see Supplementary Note 3 for a detailed discussion on the role of nonmagnetic and magnetic impurity-substrate interactions in determining the energies of the YSR states). Similar to what is observed in Fig. 2, the $d_{yz}$ state of Cr is theoretically expected at higher energy than the YSR state of $d_{z^2}$-symmetry, while for Mn the two states are found around the same energy. The calculated $d_{xz}$ state is located at lower energy than the $d_{z^2}$ state for both Cr and Mn. While this is confirmed experimentally for the $d_{xz}$ state of Mn, it was not detected for Cr, the corresponding peak being either too weak or difficult to disentangle from the adjacent dominant resonances. Note that all YSR states are characterized by a finite broadening which is related to both the experimental energy resolution and their intrinsic lifetime. This explains why, although ab initio simulations predict that each of the orbitals of Cr and Mn adatoms carries a spin moment, resulting in five distinct YSR states, not all of them are detectable experimentally, as shown in ref. [14]. Fe and V, on the other hand, are found to have a colossal adatom-substrate interactions, which is favored by the LDOS resonances located at the Fermi energy. In both cases, because of the very strong interaction for all orbitals, all YSR features are expected to appear at the edge of the SC gap, with the $d_{z^2}$ orbital dominating the scene because of its larger extension into the vacuum, which facilitates its experimental detection, in agreement with our tunneling spectra.

**Spin excitations**. The interaction of magnetic impurities with superconducting condensates is generally described within the framework of competing singlet-formation mechanisms, i.e., Kondo screening vs. Cooper pairs. This competition is captured within a phase diagram where the magnetic impurities can be either in a Kondo-screened or free-spin state depending on the impurity–superconductor coupling strength. In the strong coupling regime, $k_B T_K \gg \Delta$, with $k_B$ being the Boltzmann constant and $T_K$ the Kondo temperature, while in the weak coupling regime $k_B T_K \ll \Delta$. A quantum phase transition between these two regimes takes place for $k_B T_K \approx \Delta$, i.e., when Kondo screening and the superconducting gap are characterized by similar energies[29]. To scrutinize these aspects, a magnetic field has been applied perpendicular to the sample surface in order to quench the superconducting state. Note that all elements are characterized by a well-defined $d_{z^2}$-state, which allows to precisely map its evolution. This is found to progressively move towards the single-particle coherence peak located at the edge of the superconducting gap by increasing the orbital occupation, which should result in a progressively stronger Kondo resonance while moving from Cr to Mn and Fe. However, our measurements clearly reveal that this is far from being the case. As illustrated in Fig. 5, our data reveal a strong zero-bias anomaly (ZBA) with a step-like feature for Cr adatom, also observable in the superconducting phase as shown in Supplementary Fig. 6. Similar behavior is observed for Mn and Fe although the signal is much weaker than for Cr (see Supplementary Fig. 7 for a direct overlap of Cr, Mn, and Fe spectra) and becomes progressively broader. Finally, V and Co spectra appear totally flat.

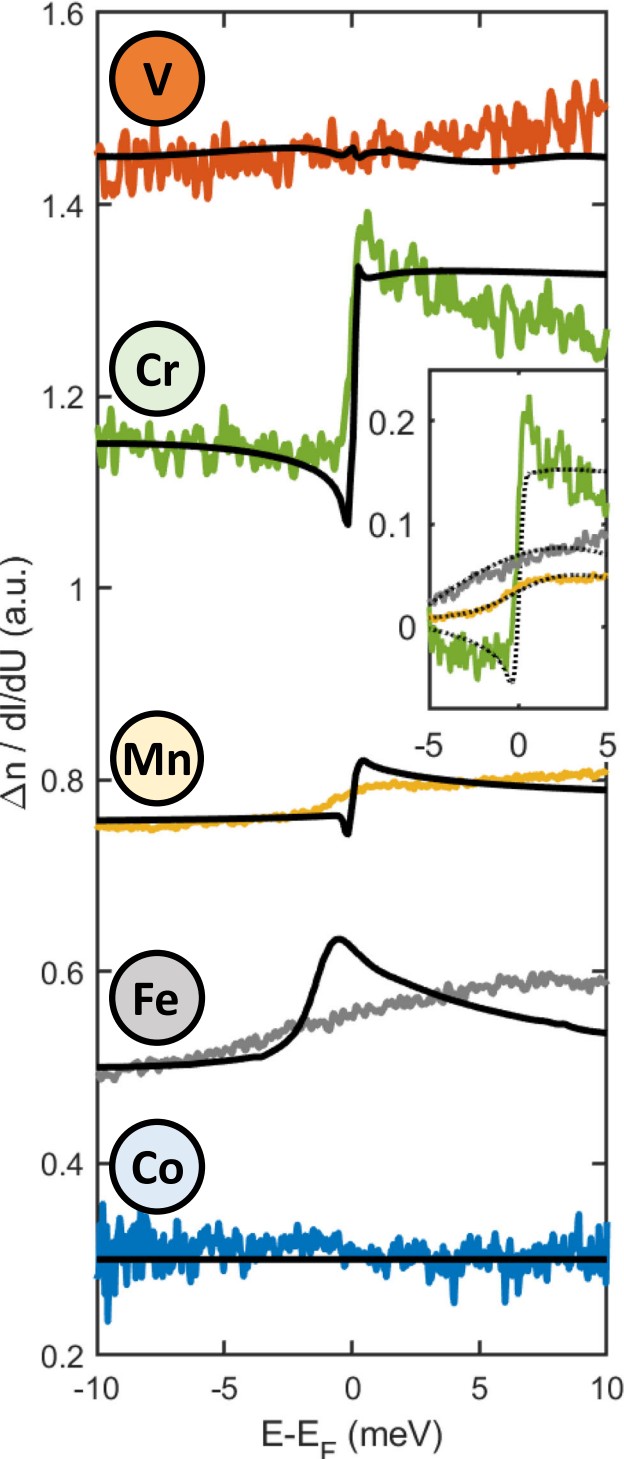

**Fig. 5 Spin-excitation spectroscopy.** For all adatoms, the experimentally obtained spectra are reported as solid lines following, for each element, the same color-coding scheme used across the manuscript. To exclude spurious effects related to the use of different microtips, all spectra are normalized to the substrate. To drive the system into a metallic state, a magnetic field has been applied perpendicular to the sample surface. All spectra are overlapped with the theoretically calculated spin-excitation spectra (solid black line). The inset shows the theoretical spectra with an artificial Gaussian broadening of 0.20 meV, 1.98 meV, and 7.78 meV for Cr, Mn, and Fe, respectively (dashed lines). Stabilization parameters: Cr, Fe, Mn: $V = 10$ meV, $I = 500$ pA; V, Co $V = 10$ meV, $I = 1$ nA.

It has recently been predicted that inelastic spin excitations can also lead to unconventional spectral shapes centered around the Fermi level[34]. To verify if this is the case, the experimental data are compared to relativistic first-principles simulations, combining TD-DFT with MBPT (see "Methods" and Supplementary Notes 4 and 5), reported as solid black lines in Fig. 5. The theoretical inelastic spectra qualitatively reproduce the experimental features (more details on the origin of the step-shapes is provided in the Supplementary Notes Notes 4 and 5) Cr has a weak MAE leading to small excitation energies. The amount of electron–hole excitations responsible for the damping of the ZBA are therefore weak, which favors the observation of the inelastic features. Electron–hole excitations are proportional to the MAE and to the product of the density of states of opposite spin-character at the Fermi energy[31,32]. Therefore, although V has a weak MAE, its small exchange splitting leads to a large LDOS at the Fermi energy and a consequent number of electron–hole excitations, heavily decreasing the lifetime of the spin excitations. The interplay of these two mechanisms, MAE and LDOS, broadens the features obtained for Mn and Fe as well. The experimental ZBA of the latter adatoms seems broader than those calculated, which can be resulting from a slight theoretical underestimation of the spin-excitation energy or of the electron–hole excitation energies as shown in Supplementary Fig. 12. Here, we account for this underestimation by broadening the theoretical spectra using a Gaussian broadening, which is shown in the inset of Fig. 5. For the three shown cases of Cr, Mn, and Fe, we used a broadening of 0.20 meV, 1.98 meV, and 7.78 meV, respectively, to match the theoretically predicted spectra with the experimental spectra.

## Discussion

Overall, our data allow to establish a unified picture of different spin-related phenomena emerging from magnetic impurities coupled to superconductors. By systematically mapping the impact of single magnetic perturbations onto the Josephson effect, we unveil the existence of a direct link between superconducting order parameter suppression and YSR states. This correlation follows a well-defined orbital occupation-dependent trend. Moreover, by comparing YSR and metallic regimes, our data challenge existing theoretical models that explain the interaction between magnetic impurities and superconductors in terms of competing for singlet-formation mechanisms, i.e., Kondo vs. Cooper pairs. Indeed, according to this picture, the asymmetry in the YSR intensity can be used to identify whenever the magnetic impurity is in a Kondo-screened ($S = 0$) or a free-spin ($S > 0$) state, with the peak intensity being stronger below and above the Fermi level, respectively. Mn and Cr are both characterized by a strong spectral weight below the Fermi level, and they are thus supposed to be in a Kondo-screened ground state ($S = 0$). In particular, we detect zero-bias anomalies which become stronger by progressively approaching the free-spin regime, indicating their unlikeliness to be Kondo resonances. Our ab initio simulations support this analysis reproducing the zero-bias anomalies by considering inelastic spin excitations. The latter hinges on the magnitude of the magnetic anisotropy energy of the adatoms. Because of the relevance of magnetic superconducting interactions in different topological qubit concepts, which lay at the foundation of advanced quantum computation architectures, the significance of our findings goes beyond the single-impurity level, evidencing that new and unexpected phases can emerge, subject to the interplay of orbital-dependent spin–substrate interactions, magnetic moments, and magnetic anisotropy energies. This can only be explored through the systematic use of a rich workbench of spectroscopy techniques for magnet-superconducting interfaces.

## Methods

**Sample and tip preparation**. Nb(110) single crystals (Surface Preparation Laboratory) have been prepared in ultra-high vacuum conditions and measured using a Tribus STM head (Scienta Omicron) operated at $T = 1.9$ K. The samples have been flashed hundreds of times at a temperature $T = 2300$ K for 12 s using an home-built electron-beam heater. As illustrated in the Supplementary Information, this procedure is necessary to progressively reduce the oxygen contamination, resulting in clean surfaces. The high quality of the surface is further confirmed by scanning tunneling spectroscopy measurements showing, in agreement with theoretical calculations, a sharp peak energetically located at $E = −0.45$ eV below the Fermi level which originates from a surface resonance of $d_{z^2}$ character. Single magnetic adatoms have deposited onto the Nb(110) surface using an electron-beam evaporator while keeping the sample at $T = 10$ K. Superconducting Nb tips have been prepared by indenting electrochemically etched W tips inside the Nb(110) for several nanometers. $dI/dU$ spectra were measured using a lock-in technique, modulating the sample bias with 50 μV (r.m.s.) ac bias at a frequency 733 Hz. More experimental details are given in Supplementary Figs. 1–7.

**Ab initio**. The ground-state properties of the adatoms deposited on Nb(110) were calculated in a two-pronged approach based on density functional theory (DFT). First, the Quantum Espresso[42,43] package was utilized for geometrical optimization of the adatom-substrate complexes. A $4 \times 4$ supercell is considered with three Nb layers and a $k$-mesh of $2 \times 2 \times 1$ is used. Exchange and correlation effects are treated in the generalized gradient approximation using the PBEsol functional[44], and we used ultrasoft pseudopotentials from the pslibrary[45] with an energy cutoff of 500 Ry. Second, the calculated positions were then used in the simulations based on the full-electron scalar-relativistic Korringa–Kohn–Rostoker (KKR) Green function including the spin–orbit interaction self-consistently[46,47]. KKR permits the embedding of single adatoms in an otherwise perfect substrate. We assume the local spin density approximation (LSDA)[48] and obtain the full charge density within the atomic sphere approximation. The angular momentum cutoff of the orbital expansion of the Green function is set to $\ell_{\max} = 3$ and a $k$-mesh of $600 \times 600$ is considered. The trend of the atomic relaxations obtained with Quantum espresso agrees with the simulations (Cr: 17%; Mn: 18%; Fe: 29%; Co: 29% of the Nb bulk interlayer distance), except for V where the theory predicts a relaxation of 22%, while from the corrugation shown in Fig. 3f, we expect a possible extra relaxation of 10%. The energies of the YSR states of the adatoms are modeled by a realistic tight-binding model with parameters from DFT. The model considers the $d$ orbitals of the adatoms and accounts for the Nb substrate via an effective Hamiltonian construction. Further details can be found in ref. [49] and Supplementary Note 2.

The spin excitations were investigated utilizing a framework based on time-dependent density functional theory (TD-DFT)[30–32] including spin–orbit interaction. Many-body effects triggered by the presence of spin excitations are approached via many-body perturbation theory[33] extended to account for relativistic effects[34]. The single-particle Green functions pertaining to the ground state are employed for the calculation of the bare Kohn–Sham dynamical magnetic susceptibility, $\underline{\chi}_{KS}(\omega)$. The latter is renormalized to $\underline{\chi}(\omega)$ via the Dyson-like equation to account for many-body effects

$$\underline{\chi}(\omega) = \underline{\chi}_{KS}(\omega) + \underline{\chi}_{KS}(\omega)\,\underline{\mathcal{K}}\,\underline{\chi}(\omega) \quad . \tag{2}$$

$\underline{\mathcal{K}}$ represents the exchange-correlation kernel, taken in adiabatic LSDA (such that this quantity is local in space and frequency-independent[50]). A magnetization sum rule permits an accurate evaluation of the energy gap in the spin-excitation spectra[30–32]. The theory was successful to describe spin excitations measured by STM (see refs. [51–53]).

The self-energy describing the interactions of the electrons and the spin excitations is calculated from a convolution of the Green function, $G$, and susceptibility, $\underline{\Sigma} \propto \underline{\mathcal{K}}\underline{\chi}\underline{G}\underline{\mathcal{K}}$ in refs. [33,34,54,55]. The impact of spin–orbit coupling is incorporated as described in ref. [34]. The self-energy is then used to renormalize the electronic structure to account for the presence of spin excitations by solving the Dyson equation $g = G + G\Sigma g$.

The theoretical spectra shown in Fig. 5 are local densities of states calculated the vacuum above the adatoms, which on the basis of the Tersoff-Hamann approach[56] correspond to the differential conductance measured by STM. More details on the simulations are provided in Supplementary Notes 1–5.

## Data availability

All data needed to evaluate the conclusions in the paper are present in the paper and/or the Supplementary Materials. Additional data related to this paper may be requested from the authors.

## Code availability

The KKR Green function code that supports the findings of this study is available from the corresponding author on reasonable request.

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

## Acknowledgements

A.M.M. and F.S.M.G. thank Juba Bouaziz for fruitful initial discussions on the metho-dology related to the description of spin excitations. S.L. acknowledges discussions with Nicolas Lorenté. A.M.M., F.S.M.G., S.B., and S.L. are supported by the European Research Council (ERC) under the European Union's Horizon 2020 research and

innovation program (ERC-consolidator grant 681405 DYNASORE). We acknowledge the computing time granted by the JARA-HPC Vergabegremium and VSR commission on the supercomputer JURECA[57] at Forschungszentrum Jülich and at the supercomputing center of RWTH Aachen University.

## Author contributions

F.K. performed the STM measurements and analyzed the data. A.M.M. and S.B. performed the ab initio simulations. S.B. and S.L. conceived the theoretical framework describing the YSR states. A.M.M., F.S.M.G., S.B., and S.L. analyzed the theoretical data. S.L. and P.S. wrote the initial version of the paper to which all authors contributed. S.L., S.S.P.P., and P.S. supervised the project.

## Funding

## Competing interests

The authors declare no competing interests.
