## [Peer Review File · Nature Communications]

REVIEWER COMMENTS

Reviewer #1 (Remarks to the Author):

The paper investigates V, Cr, Mn, Fe, and Co atoms on a superconducting Nb substrate. Scanning tunneling spectroscopy is used to resolve YSR states and the peak height in a voltage-biased Josephson junction. In addition, DFT calculations are employed to understand the experimental observations. Unfortunately, experimental results and simulations do not fit very well and therefore do not provide new understanding of YSR states, Josephson effect and spin excitations, which would justify publication in Nature Communications.

1. The STS spectra show three YSR states for Cr and Mn on Nb. On the contrary, theory predicts five states. Their theoretical energy separation suggests that all of them should be well resolved in experiment. In addition, not even the energetic sequence of the YSR states from the individual d orbitals agrees with experiment. Why does theory only give the absolute value of the YSR energy? Theory should be able to tell whether the spin is in the Kondo screened or free spin state. An immediate question is if the adsorption site in experiment is the one simulated in theory. Is there only one adsorption site in experiment?
2. Josephson spectroscopy is used to measure reduction of the superconducting order parameter. The authors claim that one can deduce the superconducting order parameter quantitatively. Hence, by how much is the order parameter reduced? What are the predictions from theory and how do they compare with experiment?
3. Experimental parameters are lacking throughout the paper. This is particularly important for Josephson spectroscopy. The junction resistances need to be given for each spectrum.
4. I do not see a step on Mn and Fe atoms in figure 5, while theory predicts one. I do not agree with the authors' interpretation that the "theoretical spectra in figure 5 nicely reproduce the experimental features".
5. Some of the authors already conducted a similar study investigating the evolution of YSR states upon orbital filling on a Re surface (Ref.38). The results should be discussed and compared.

Reviewer #2 (Remarks to the Author):

This manuscript details the magnetic properties of a series of transition metal adatoms on a Nb substrate in both superconducting and metallic regimes. Interpretation of the experimental data is supported by substantial theoretical calculations. The authors show the presence of multiple YSR states, map their spatial and spectroscopic fingerprints, and demonstrate that magnetism suppresses the Josephson effect. Similar results have all been reported in past works, but not as systematically documented in a single paper. The authors also measure spectra in a large parallel field and ascribe their shape to inelastic spin excitations. Here, the data are more novel but the match to theory is less clear. All results are interpreted in the context of spin-superconductor interaction strength, magnetic anisotropy, and spin excitations; the key conclusion is that the Kondo vs. singlet paradigm appears to be insufficient to explain all the data.

The authors should address the following.

-In Fig. 5, the experimental data do not qualitatively match the theoretical simulations as claimed. In particular, the data for Mn and Fe do not match the theoretical lineshape; they are comparably featureless as the "totally flat" V and Co data. Therefore, the conclusion that inelastic spin excitations explain all the data is not well supported.

-Relatedly, the axis labels on the left of Fig. 5 are AU without the zero labeled. Therefore, it is impossible to tell whether this is a large feature relative to the background (even for Cr) or not. This should be fixed.

-Can the authors comment to what extent the application of a relatively large magnetic field itself affects the spin dynamics? Have any other measurements been performed at different parallel magnetic fields?

-As far as I can tell, the authors present data from just one atom of each type. To what degree is this behavior reproducible? Further examples should be shown in the supplement. For example, are all magnetic adatoms always in the same lattice position relative to the Nb? This is especially a worry since magnetic atom evaporation is done cold, so atom mobility may be limited.

-For clarity in methods and reproduction, the authors should mention somewhere the tunneling parameters, e.g. setpoint current and bias.

REVIEWER COMMENTS

Reviewer #1 (Remarks to the Author):

The paper investigates V, Cr, Mn, Fe, and Co atoms on a superconducting Nb substrate. Scanning tunneling spectroscopy is used to resolve YSR states and the peak height in a voltage-biased Josephson junction. In addition, DFT calculations are employed to understand the experimental observations. Unfortunately, experimental results and simulations do not fit very well and therefore do not provide new understanding of YSR states, Josephson effect and spin excitations, which would justify publication in Nature Communications.

Answer: We thank the referee for his/her critical comments on our manuscript, which motivated us to significantly improve it (see modified text shown in red in the main text and supplementary information). As described in the following, we took all referee's comments very seriously. This has been done by simultaneously addressing several aspects, in particular:

- (i) collecting and presenting more experimental data;
- (ii) providing a more in-depth analysis of our findings;
- (iii) reformulating some aspects of our work, which were not sufficiently clear in the first version of the manuscript.

We would like to stress that our work is the first systematic investigation of spin-superconductor interactions by spanning the family of 3d atoms utilizing three different but complementary spectroscopic techniques, which enabled us to investigate different tunneling regimes: YSR, Josephson, and metallic. Our workbench of combined experimental techniques produces a large ensemble of data permitting us to evidence for the first time that:

- (i) the framework generally used to describe magnetic impurities interacting with superconducting condensates is not sufficient to correctly explain the delicate interplay between magnetic moments and Cooper pairs. As recognized by Referee # 2, this is the "*the key conclusion*" of our study. As explained in the main text, this was not experimentally scrutinized so far, mainly because the overwhelming majority of previous studies focused on just a single type of impurity, either atoms or molecules, thus preventing the discovery of new trends, as we do in our study. This becomes evident by systematically correlating Fig. 2 (YSR states) and Fig. 5 (metallic regime) of the main manuscript.

Beside this important aspect, our work contains two more additional major elements of novelty, namely:

- (ii) it allows to establish a direct link between the suppression of the superconducting order parameter and the presence of multiple Shiba states residing within the superconducting gap. This becomes evident by systematically correlating Fig. 2 (YSR states) and Fig. 3 (Josephson spectroscopy).;
- (iii) it allows to shed new light on fundamental low energy excitations related to the spin-degree of freedom, and to directly link them to fundamental quantities such as the orbital occupation and the magnetic anisotropy energy.

Our work is predominantly experimental evidence-driven. Backed up by the simulations, it results in a compelling and self-consistent picture. The discovery of well-defined trends and correlations is found for all elements and in different spectroscopic modes (comparison between Fig.2, Fig. 3., and Fig.5).

However, the referee comments motivated us to perform new measurements and analysis. We believe that the agreement between experiments and theory is now pretty remarkable, the energy scale at play being very small, i.e. in the meV regime and below. Additionally, the theoretical results are obtained independently from the experiment, i.e. we do not perform a

fit! The parameters used in the model describing the superconducting-magnetism physics come fully from ab-initio. The overall agreement between the experiments and theory has been significantly improved, and the same trends are now consistently visualized, both in our experimental data as well as in our calculations, as elaborated below.

- (i) *Shiba states*: the d_{z^2} orbital, which can be effectively mapped in all elements and which lies at the heart of the discovered breakdown of the commonly used picture based on competing singlet formation mechanism, i.e. Kondo vs. Cooper pair (comparison of Figure 2 and Figure 5), is found to be perfectly consistent between theory and experiments.
- (ii) *Josephson spectroscopy*: The same very good theory-experiment agreement is also found when analyzing the discovered trend in suppressing the Josephson supercurrent, reported in Fig. 3. Here, the suppression becomes progressively stronger in the following order: Co, V, Fe, Mn, and Cr, perfectly matching the trend of the calculated magnetic moments: for the adatom/substrate ensemble: Co = $0 \mu_B$, V = $1.4 \mu_B$, Fe = $1.5 \mu_B$, Mn = $3.0 \mu_B$, Cr = $3.3 \mu_B$.
- (iii) *Spin-excitations*: An excellent agreement is achieved for 4 elements over 5. The feature detected experimentally on Fe seems larger than the theoretical one (explanation given below and in the main text). However, the increased broadening while progressively moving from Cr, Mn, and Fe, is consistently found in experiments as well as in our theoretical calculations.

A more detailed answer can be found in the point-by-point response below.

1. (i) The STS spectra show three YSR states for Cr and Mn on Nb. On the contrary, theory predicts five states. Their theoretical energy separation suggests that all of them should be well resolved in experiment. (ii) In addition, not even the energetic sequence of the YSR states from the individual d orbitals agrees with experiment. (iii) Why does theory only give the absolute value of the YSR energy? (iv) Theory should be able to tell whether the spin is in the Kondo screened or free spin state. (v) An immediate question is if the adsorption site in experiment is the one simulated in theory. Is there only one adsorption site in experiment?

Answer: We thank the Reviewer for raising the various questions mainly focused on the comparison of the experimental-theoretical YSR spectra, which allowed us to improve/complete the related text.

- (i) The ab-initio simulations predict that each of the orbitals of Cr and Mn adatoms carries a spin moment. Therefore 5 YSR states are expected but not all of them are necessarily detectable experimentally, (as also found previous works, see for example the seminal work PRL 100,226801, the first high resolution study of magnetic impurities-induced bound states in superconductors, where three YSR signatures are detected for Cr and only two for Mn although being experiments performed at a temperatures significantly lower than ours). This is because STS is a technique that permits to detect the electronic states of the adsorbate-substrate complex that have the right symmetry to decay into the vacuum and overlap with those of the tip. Within the Tersoff-Hamann model, the tip carrying an s-orbital, would only select the s, p_z and d_{z^2} orbitals. The rest of the states would be silent. Since the tip is not simply carrying an s-orbital, more states could in principle be observed. This explains the prominence of the d_{z^2} in the experimental STS spectra. Thus, the analysis/conclusions performed along the manuscript are driven by the dominant d_{z^2} state.
- (ii) The model used to predict the energy of the YSR states, although taking input from ab-initio results, is based on various assumptions, which are better listed and

elaborated in the new version of the manuscript/supplement. Currently, there is no full ab-initio theory/framework for superconductors interacting with magnets. First, the model is based on a mean field approach without self-consistency. It is known that this affects the position of YSR states. Second, the s—d coupling, magnetic J and non-magnetic V, are extracted from ab-initio on the basis of the Schrieffer-Wolff transformation, which is valid for the case of a single electron interacting weakly with the electronic bath characterized by a constant density of states. Third, the quantities entering the definition of J and V are hybridizations functions, exchange splitting and crystal field which are of the order of eV, leading to a delicate balance/competition of the magnetic and non-magnetic scattering contributions resulting in YSR energies of the order of meV and sub-meV.

Considering all of these aspects, we find that the trend in the location of the theoretical d_z^2 YSR state is reasonable when compared to that of the experiment. For instance, the one of Cr is predicted to be located at a lower energy than that of Mn. Note that if the non-magnetic adatom-substrate interaction was not included, that state would be located at almost the same energy (see Supplementary Note 10 for a detailed description). Similarly to what is observed in Fig.2, the d_{yz} state of Cr is theoretically expected at a higher energy than the YSR state of d_z^2 -symmetry, while for Mn the two states are found around the same energy. The calculated d_{xz} state is located at a lower energy than the d_z^2 state for both Cr and Mn. While this is confirmed experimentally for the d_{xz} -state of Mn, it was not detected for Cr, the corresponding peak being either too weak or difficult to disentangle from the adjacent dominant resonances. Note that all YSR states are characterized by a finite broadening which is related to both the experimental energy resolution and their intrinsic lifetime. The detection aspect has been addressed in (i), which becomes extremely difficult if the intensity of the peak is very weak or if located at the vicinity of another but dominant peak. Fe and V, on the other hand, are found to have a colossal adatom-substrate interaction (listed in Supplementary Table S8), which is favored by the LDOS resonances located at the Fermi energy. In both cases, because of the very strong interaction for all orbitals, all YSR features are expected to appear at the edge of the SC gap, with the d_z^2 orbital dominating the scene because of its larger extension into the vacuum, which facilitates its experimental detection, in agreement with our tunneling spectra. We note that it would be possible to change one of the quantities entering the adatom-substrate interaction to tune the position of the YSR states to those obtained experimentally. But we refrain from doing that since the major trends are reasonably reproduced.

- (iii) As well known, the YSR states come in pairs at positive and negative energies (with respect to the Fermi energy). Therefore, we give the absolute values.
- (iv) The strength of the adatom-substrate interactions quantified in terms of alpha (magnetic) and beta (non-magnetic) are listed in the Supplementary Table S8. Strong versus small coupling can be identified when alpha crosses 1 [*Rev. Mod. Phys.* 78, 373–433 (2006)], if one limits the analysis to the magnetic interaction only. The threshold is modified owing to the presence of the non-magnetic interaction. However, it is well known that the threshold is reduced if self-consistency is adopted [*Phys. Rev. B* 55, 12648 (1997)]. As analyzed in Supplementary Note 10, all Cr and Mn orbitals couple antiferromagnetically to the substrate's states in contrast to Fe orbitals, for which only the d_z^2 and d_{xz} are antiferromagnetic. The magnetic and non-magnetic coupling are found generally rather strong reaching almost 1 or larger, with the largest interactions characterizing Fe because of the electronic states being close to the Fermi energy. The weakest interacting orbitals are the $d_{x^2-y^2}$ states of Cr and Mn and the d_{yz} state of Cr. The rest are found strongly interacting.

- (v) Yes, there is only one adsorption site for all the adatoms, with each element showing a very distinct behavior. This information has now been included in the Supplementary Note 4 (and reported here for convenience) by showing 5 different STS spectra for each element, acquired in different sample preparations and also by using different tips. This proves the robustness and reproducibility of our experimental results. Slight differences in the energy position are due to the use of different STM tips, characterized by slightly different superconducting gaps.

Figure 1: Spectroscopic data for different adatoms. For each 3d element, the data are representative of measurements acquired on different sample preparations as well as by using different superconducting tips. For each 3d element, all adatoms always show the same behaviour. The energy position of the Shiba states depends on adatom-substrate hybridization. The high reproducibility of the data confirms and further supports the existence of a single adsorption site. Small differences in the absolute energy position of the Shiba peaks (orange curves) are related to the use of different Nb-coated tips.

Adatoms are found to adsorb in the hollow site of the Nb(110). This is also consistent with our theoretical calculations, which predict the hollow site as being the most energetically favorable site. The determination of the adsorption site is very challenging in STM, since atomically resolved images at metallic surfaces are generally obtained at very small voltages and very high current, i.e. small tunneling resistance. In this case, STM corrugation is essentially a contour map of constant surface local density of states $\rho(r, E_F)$ with r being the tip position and E_F the Fermi level, as explained in the seminal work of Tersoff and Hamann (PRL 50, 1998 and PRB 31, 805). In their work, it was already anticipated that orbital symmetries different from s-wave (not only in the sample, but also in the probing tip) can significantly impact the measured corrugation. This can also give rise to corrugation reversal effects, as later rationalized by Chen in PRL 69, 1656. As found by Heinze and co-workers, these effects are expected to be particularly strong in the case of (110) surfaces, where highly directional d orbitals dominate the scene close to the Fermi level, especially in the case of Nb(110). As described in Phys. Rev. B 58, 16432 “The competition between surface resonances and surface states is a quite general mechanism and anticorrugation is expected to occur on (110) surfaces of other bcc transition-metals [i.e., Nb(110), Mo(110), Ta(110)].”. To the best of our knowledge, this effect always escaped experimental verification on Nb(110), but we could clearly detect it in our work, as visualized in the image below, now added as Supplementary Note 3.

Figure 2: All adatoms are characterized by a single and the same adsorption site, which is found to be the hollow site of the Nb(110) surface.

All adatoms are characterized by a single and the same adsorption site: the hollow site, which is the most energetically favorable and used in our calculations [for the case of Cr (Fe), the hollow site is more preferable than the top site by 0.62 eV (1 eV). The bridge site is also unfavorable by 0.25 eV (0.35 eV) with respect to the hollow site]. However, depending on the exact tunneling conditions, i.e. tip orbitals and tunneling parameters, the atoms can appear anticorrugated, i.e. top looking as hollow and hollow as top. This is demonstrated for Mn and Fe (panel b). Note that this is just how atoms appear, and it is not related to possible different positions. This is confirmed by STS measurements, which are fully reproducible independently from the atom appearance, further corroborating the existence of a single adsorption site.

2. Josephson spectroscopy is used to measure reduction of the superconducting order parameter. The authors claim that one can deduce the superconducting order parameter

quantitatively. Hence, by how much is the order parameter reduced? What are the predictions from theory and how do they compare with experiment?

Answer: We would like to stress that we never claimed that one can deduce the superconducting order parameter quantitatively. However, it is well established that Josephson current is affected by the superconducting order parameter. Hence, we can once more look at trends, which is at the core of our work. A nice thorough Josephson-based investigation by Ast et al. [NatComm 7,13009] demonstrated that non-linear contributions to the critical currents depends on other contributions. In their case, even a non-magnetic defect triggers changes in the current of the order of 6% with respect to that obtained in the free-defect region. Assuming that such effects can reach 10%, which defines an approximate error bar, we can safely state that the 62% and 30 % reduction observed in the dI/dU peak for Cr and Mn, respectively, followed by 18% for Fe are direct signatures of a stronger renormalization of the superconducting order parameter for Cr and Mn in comparison to that induced by Fe.

Theoretically, with a non-self-consistent scheme, we cannot address the renormalization of the superconducting parameter. It is, however, expected that the superconducting order parameter decreases at the vicinity of magnetic impurities proportionally to the magnetic adatom-substrate interaction (see e.g. [Phys. Rev. B **92**, 064503]). Of course, stronger magnetic interactions would trigger a stronger renormalization but with a more complicated fashion than a linear dependence. Also, one has to keep in mind that we are dealing with multi-orbital cases interacting magnetically and non-magnetically with the substrate, which defines a formidable and exciting challenge to tackle in the future. We expect various competing contributions to the overall critical current. So, if one takes the magnetic interaction of the d_{z^2} orbital, we would expect that Fe should lead to a stronger renormalization of the order parameter with respect to that of Cr and Mn. However, the latter elements host more orbital (all five) coupled antiferromagnetically to the substrate, which explains the correlation of the experimental spectra with the magnitude of the adatoms moments.

3. Experimental parameters are lacking throughout the paper. This is particularly important for Josephson spectroscopy. The junction resistances need to be given for each spectrum.

Answer: We fully agree with the Referee that junction resistances are very important. These parameters have now been added in the revised version of the manuscript.

4. I do not see a step on Mn and Fe atoms in figure 5, while theory predicts one. I do not agree with the authors' interpretation that the "theoretical spectra in figure 5 nicely reproduce the experimental features".

Answer: In the measurements displayed in the initial version of the manuscript, a magnetic field had to be applied to kill superconductivity on both the tip and substrate. We repeated the experiment with a metallic tip, which allowed us to apply a smaller magnetic field to reach the metallic regime. The new spectra are shown below and in the new manuscript (Fig.5). One sees that zero-bias features are observed for the three adatoms: Cr, Mn and Fe. The broadening is the largest for Fe, followed by Mn then Cr. Although the energy scale of the observed features is in the meV range, the systematic comparison with the ab-initio predictions is impressive for the 5 elements: V, Cr, Mn, Fe and Co atoms. The theoretical technique has recently recovered the zero-bias anomalies of Co atoms on Cu, Ag, and Au(111) surfaces also located in the meV range [arXiv:2003.01746]. The physics behind the spectra anomalies is dictated by the magnetic anisotropy energy, i.e. it is a relativistic effect, which opens a gap in the spin-excitations spectra with a broadening (lifetime) shaped by the electron-hole excitations of opposite spin-character. As discussed in the main text, the electron-hole

excitations are given approximately by the product of the density of states of opposite spin-character, and increase linearly with the spin-excitations energy (proportional to the magnetic anisotropy energy). If the experimental magnetic anisotropy energy theory underestimates the magnitude of the magnetic anisotropy energy, the experimental spectra would look broader than theoretical ones (see Figure below and Supplementary Figure 12).

Figure 3 Overlap of Cr, Mn, and Fe experimental spectroscopic data in the metallic regime. A clear step-like feature is visible for Cr, with an intensity which is significantly stronger than those observed for the other atomic species.

Figure 4: Spin excitation spectroscopy. For all adatoms, the experimentally obtained spectra are reported as solid lines following, for each element, the same color coding scheme used across the manuscript. To drive the system into a metallic state, a magnetic field has been applied perpendicular to the sample surface. All spectra are overlapped with the theoretically calculated spin-excitation spectra (solid black line). A good qualitative agreement is obtained for all elements. Stabilization parameters: Cr, Fe, Mn: $V = 10$ meV, $I = 500$ pA; V, Co $V = 10$ meV, $I = 1$ nA.

Figure 5: Impact of the resonance position and its broadening on the local density of states. a) Fit of the minority and majority spin channel of the theoretical renormalized density of states of the Fe. b) The total density of states as function of the resonance position E_0 and its broadening Γ . The impact of an underestimated magnetic anisotropy is simulated by increasing simultaneously E_0 and Γ . Shown are three different parameter sets with the original fitted parameters (blue curve) and a doubling of E_0 and Γ (red curve) as well as a tripling of E_0 and Γ (green curve).

5. Some of the authors already conducted a similar study investigating the evolution of YSR states upon orbital filling on a Re surface (Ref.38). The results should be discussed and compared.

Answer: We thank the reviewer for pointing to the work reported by one of us in Ref.38. The two studies are distinct since ours targets a larger number of transition elements: V, Cr, Mn, Fe and Co adatoms, while that on Ref.38 was devoted to Mn, Fe and Co adatoms. A crucial difference is the large gap characterizing Nb, which enables to resolve more YSR states than on the Re surface. Interestingly, on both surfaces, Re or Nb, Co can lose its magnetic moment. Moreover, our study is based on a systematic Josephson spectroscopy, which as we show provides crucial information not accessible with other STS techniques. Theoretically, the ab-initio technique utilized in the present paper is not available in Ref.38, where the experimental spectra were not reproduced from first-principles. Finally, the trends found in Re were explained in terms of the conventional picture in contrast to what we find on Nb.

Following the recommendation of the Referee, we added more text related to the comparison with Ref.38: “Similarly to our finding, Co adatoms can be non-magnetic on the Re(0001) surface as revealed by a YSR study limited to Mn, Fe and Co impurities”.

Reviewer #2 (Remarks to the Author):

This manuscript details the magnetic properties of a series of transition metal adatoms on a Nb substrate in both superconducting and metallic regimes. Interpretation of the experimental data is supported by substantial theoretical calculations. The authors show the presence of multiple YSR states, map their spatial and spectroscopic fingerprints, and demonstrate that magnetism suppresses the Josephson effect. Similar results have all been reported in past works, but not as systematically documented in a single paper. The authors also measure spectra in a large parallel field and ascribe their shape to inelastic spin excitations. Here, the data are more novel but the match to theory is less clear. All results are interpreted in the context of spin-superconductor interaction strength, magnetic anisotropy, and spin excitations; the key conclusion is that the Kondo vs. singlet paradigm appears to be insufficient to explain all the data.

Answer: We thank the Referee for a careful reading of our manuscript and for his/her feedback. Here we address the concerns raised, which in general are related to the comparison between theory and experiment. The text modified in the main manuscript and supplementary information as well as the additional material is highlighted in red. We would like to stress that, in addition to the breakdown of the Kondo vs. singlet paradigm, an additional major aspect of novelty of our work is the direct correlation of Josephson supercurrent suppression with the number of YSR states, a trend which is consistently found in line with the magnetic moments as inferred by ab-initio calculations. In particular the current suppression becomes progressively stronger in the following order: Co, V, Fe, Mn, and Cr, perfectly matching the trend of the calculated magnetic moments: for the adatom/substrate ensemble: $\text{Co} = 0 \mu_B$, $\text{V} = 1.4 \mu_B$, $\text{Fe} = 1.5 \mu_B$, $\text{Mn} = 3 \mu_B$, $\text{Cr} = 3.3 \mu_B$.

The authors should address the following.

1) In Fig. 5, the experimental data do not qualitatively match the theoretical simulations as claimed. In particular, the data for Mn and Fe do not match the theoretical lineshape; they are comparably featureless as the “totally flat” V and Co data. Therefore, the conclusion that inelastic spin excitations explain all the data is not well supported.

Answer: We refer to our answer to question 4 of Referee #1. The new data show a very reasonable agreement between the ab-initio inelastic spectra and those obtained experimentally.

2) Relatedly, the axis labels on the left of Fig. 5 are AU without the zero labeled. Therefore, it is impossible to tell whether this is a large feature relative to the background (even for Cr) or not. This should be fixed.

Answer: We apologize for the missing labels in the first version of the manuscript. These have now been included. Additionally, a direct overlap of Cr, Mn, and Fe has now been added as Supplementary Note 7 to better verify this aspect.

3) Can the authors comment to what extent the application of a relatively large magnetic field itself affects the spin dynamics? Have any other measurements been performed at different parallel magnetic fields?

Answer: Measurements have been performed at different magnetic fields applied perpendicular to the sample surface. A clear trend is visible for Cr, whose sharp transition at the Fermi level allows one to effectively monitor the spectra evolution as a function of the

magnetic field for the maximum field available in our set-up (5T). The details of these unconventional spin-excitation will be addressed in a separate dedicated study under preparation.

Figure 6: Redacted

Additionally, the spin excitation for Cr remains well visible and coexisting within the SC regime. To the best of our knowledge, it is the first time that spin excitations are found to coexist within the superconducting gap. This information has now been included in the Supplementary Note 6 (see also below).

Figure 7: Scanning tunneling spectroscopy measurements on Cr adatom acquired using superconducting tip. The high intensity of the spin-excitation for Cr compared to the other elements (See Fig. 5 in the main text), makes it possible to clearly visualize its coexistence when the sample is superconducting, as visualized in (a). The emergence of a step function stays well visible when the Nb(110) substrate is turned in a normal metallic regime by applying a strong out-of-plane magnetic field (b). The residual gap visible in the spectrum corresponds to the tip superconducting gap, whose superconducting cluster at the apex is characterized by a much higher critical field than the sample due to its finite size.

4) As far as I can tell, the authors present data from just one atom of each type. To what degree is this behavior reproducible? Further examples should be shown in the supplement. For example, are all magnetic adatoms always in the same lattice position relative to the Nb? This is especially a worry since magnetic atom evaporation is done cold, so atom mobility may be limited.

Answer: We thank the referee for his/her comment. For each element, several atoms have been measured and all showed the same behavior. To address this important point raised by the referee, new data have been now included in the Supplementary Note 4.

5) For clarity in methods and reproduction, the authors should mention somewhere the tunneling parameters, e.g. setpoint current and bias.

Answer: The tunneling parameter have been now included in the manuscript.

REVIEWER COMMENTS

Reviewer #1 (Remarks to the Author):

The authors have submitted a significantly improved manuscript. However, dI/dU spectra in Fig. 5 are still not very convincing. The inelastic excitations are not well visible and comparison to theory rather weak. I suggest that authors add background spectra to the figures, such that excitations may at least be visible in comparison to the clean substrate.

Moreover, despite of YSR states appearing in pairs, calculated energies should carry a sign and provide information on the ground state of the system.

Reviewer #2 (Remarks to the Author):

The authors have adequately addressed nearly all of my technical questions in their revised manuscript, which is significantly improved. However, one critical issue - agreement between theoretical and experimental spectra - is still lacking. In the new data (Fig. 5), there is a more pronounced broad feature in the spectra for Mn and Fe, but I still don't see a good correspondence between theory and experiment for either. I do not agree that 'the theoretical inelastic spectra qualitatively reproduce the experimental features' for these cases. The authors argue that enhanced broadening, due to differences in resonance energy or broadening can smear out the expected features. However, even increasing these threefold (which the authors call a 'slight theoretical underestimation') does not smear out the theoretical curves as much as the experimental ones. With this remaining discrepancy, it is hard to have confidence in the theoretical interpretation. At the very least, the authors should try to actually fit the experimental curves and see what approximate ranges of resonance and broadening parameters would be needed within this theoretical construct. Although I believe the experimental work is sound and comprehensive, and the above question is the only remaining outstanding issue in my view, it is sufficiently important to the key conclusions that I still cannot recommend publication of the manuscript in its current form.

RESPONSE TO REFEREES

Reviewer #1 (Remarks to the Author):

The authors have submitted a significantly improved manuscript. However, dI/dU spectra in Fig. 5 are still not very convincing. The inelastic excitations are not well visible and comparison to theory rather weak. I suggest that authors add background spectra to the figures, such that excitations may at least be visible in comparison to the clean substrate. Moreover, despite of YSR states appearing in pairs, calculated energies should carry a sign and provide information on the ground state of the system.

Reviewer #2 (Remarks to the Author):

The authors have adequately addressed nearly all of my technical questions in their revised manuscript, which is significantly improved. However, one critical issue - agreement between theoretical and experimental spectra - is still lacking. In the new data (Fig. 5), there is a more pronounced broad feature in the spectra for Mn and Fe, but I still don't see a good correspondence between theory and experiment for either. I do not agree that 'the theoretical inelastic spectra qualitatively reproduce the experimental features' for these cases. The authors argue that enhanced broadening, due to differences in resonance energy or broadening can smear out the expected features. However, even increasing these threefold (which the authors call a 'slight theoretical underestimation') does not smear out the theoretical curves as much as the experimental ones. With this remaining discrepancy, it is hard to have confidence in the theoretical interpretation. At the very least, the authors should try to actually fit the experimental curves and see what approximate ranges of resonance and broadening parameters would be needed within this theoretical construct. Although I believe the experimental work is sound and comprehensive, and the above question is the only remaining outstanding issue in my view, it is sufficiently important to the key conclusions that I still cannot recommend publication of the manuscript in its current form.

Answer:

We thank both Reviewers for carefully reading our response and acknowledging our efforts in addressing their points of concern. We are pleased to read that both Referees find the manuscript significantly improved. Their only left criticism relates to the comparison between theoretical and experimental spectra for Fe and Mn in Figure 5.

We would like to stress once again that our theoretical analysis of the experimental signatures emerging at the Fermi for V, Cr, Mn, Fe, and Co is not based on simplified models but on fully ab-initio calculations. In this context, the small energy scales at play require to push the resolution in the meV range and below, which is at the limit of what is nowadays possible in state-of-the-art density functional theory.

We very much appreciate the suggestions of both referees to improve the agreement for Fe and Mn, and follow the advice of Referee 2, which takes care of the request of Referee 1. As described in the manuscript, electron-hole excitations are responsible for the broadening of the inelastic features. Now, we identified the amount of broadening underestimated theoretically that is required to reasonably fit the experimental inelastic features: 0.20 meV, 1.98 meV and 7.78 meV for Cr, Mn and Fe, respectively. The spectra with their respective theoretical fits (dashed black lines) are reported below and they have now been included as in the inset amended to Fig.5 in the main text.

To describe the fitting procedure, the following sentences have been added to the main text:

“Here we account for this underestimation by broadening the theoretical spectra using a Gaussian broadening, which is shown in the inset of Figure 5. For the three shown cases of Cr, Mn, and Fe we used a broadening of 0.20 meV, 1.98 meV and 7.78 meV, respectively, to match the theoretically predicted spectra with the experimental spectra.”

Moreover, by directly overlapping the spectra, the inset also better highlights the features detected at the Fermi for Fe and Mn, allowing to compare their intensity to that of Cr. To improve the readability of the figure, we did not include the spectra obtained on V and Co which, in agreement with our calculations, are featureless. Note that, as described in the caption, all spectra are normalized to the bare Nb substrate. This guarantees that the spectroscopic signatures obtained for all adatoms are real and not affected by spurious variations related to the use of different microtips.

We are glad to read that Referee 2 already clearly states that “the experimental work is sound and comprehensive”. We believe our response addresses the comments of the referees and we would like to thank them once again for their constructive criticism which helped to improve the quality of our work.